# Hepatocyte Nuclear Factor 1α Proinflammatory Effect Linked to the Overexpression of Liver Nuclear Factor–κB in Experimental Model of Chronic Kidney Disease

**DOI:** 10.3390/ijms23168883

**Published:** 2022-08-10

**Authors:** Elzbieta Sucajtys-Szulc, Alicja Debska-Slizien, Boleslaw Rutkowski, Marek Szolkiewicz, Julian Swierczynski, Ryszard Tomasz Smolenski

**Affiliations:** 1Department of Nephrology, Transplantology, and Internal Medicine, Medical University of Gdansk, Smoluchowskiego 17, 80-214 Gdansk, Poland; 2Department of Cardiology and Interventional Angiology, Kashubian Center for Heart and Vascular Diseases in Wejherowo, Pomeranian Hospitals, 84-200 Wejherowo, Poland; 3Koszalin State Higher Vocational School, Lesna 1, 75-582 Koszalin, Poland; 4Department of Biochemistry, Medical University of Gdansk, Debinki 1, 80-211 Gdansk, Poland

**Keywords:** hepatocyte nuclear factor 1α, nuclear factor–κB, interleukin-6, vascular cell adhesion molecule 1, intercellular cell adhesion molecule 1, monocyte chemoattractant protein 1, clofibrate, chronic kidney disease

## Abstract

Chronic kidney disease (CKD) is associated with low-grade inflammation that activates nuclear factor–κB (NF–κB), which upregulates the expression of numerous NF–κB responsive genes, including the genes encoding IL-6, ICAM-1, VCAM-1, and MCP-1. Herein, we found the coordinated overexpression of genes encoding RelA/p65 (a subunit of NF–κB) and HNF1α in the livers of chronic renal failure (CRF) rats—an experimental model of CKD. The coordinated overexpression of *RelA/p65* and *HNF1α* was associated with a significant increase in *IL-6*, *ICAM-1*, *VCAM-1*, and *MCP-1* gene expressions. A positive correlation between liver RelA/p65 mRNA levels and a serum concentration of creatinine and BUN suggest that *RelA/p65* gene transcription is tightly related to the progression of renal failure. The knockdown of *HNF1*α in the HepG2 cell line by siRNA led to a decrease in Rel A/p65 mRNA levels. This was associated with a decrease in *IL-6*, *ICAM-1*, *VCAM-1*, and *MCP-1* gene expressions. The simultaneous repression of HNF-1α and RelA/p65 by clofibrate is tightly associated with the downregulation of *IL-6*, *ICAM-1*, *VCAM-1*, and *MCP-1* gene expression. In conclusion, our findings suggest that NF–κB could be a downstream component of the HNF1α-initiated signaling pathway in the livers of CRF rats.

## 1. Introduction

Chronic kidney disease (CKD) is associated with low-grade inflammation and increases the serum concentration of inflammatory markers, such as C-reactive protein, cytokines, TNF α, and adhesion molecules [1,2,3]. It has also been proposed that inflammation might promote renal CKD progression [2].

It is generally believed that inflammation activates nuclear factor–κB (NF–κB), a family of transcription factors that, in turn, upregulates the expression of several genes encoding inflammatory mediators [4,5,6]. An NF–κB family comprises five members, including Rel A/p65, Rel B, c-Rel, NF–κB 1 (p50), and NF–κB 2 (p52), which form a different array of homo- or heterodimers. The most prominent dimer of NF–κB is the 65/50 heterodimer, which stays inactivated in the cytosol of unstimulated cells (i.e., under physiological conditions) by binding to inhibitory proteins (I-κBα, I-κB β, or I-κB) [6]. I-κB proteins’ site-specific phosphorylation, which is catalyzed by a multi-subunit I-κB kinase complex and rapid degradation through the ubiquitin–proteasome pathway, results in the release and translocation of an active form of NF–κB (including Rel A/p65) to the nucleus. There, it binds to κB DNA sites with the consequent activation of target genes encoding: (a) inflammatory response proteins (such as cytokine: IL-6) [4,5,6]; (b) adhesion molecules (such as ICAM-1 and VCAM-1); and (c) chemokines (MCP-1) [4,5,6]. This is a primary mechanism for regulating the NF–κB pathway. However, numerous papers have reported that the NF–κB signaling pathway might also be regulated at the transcriptional level, and this upregulation is related to some pathologies [7,8,9,10,11,12,13,14,15]. 

Hepatocyte nuclear factor 1α (HNF1α) is a transcriptional factor that plays several roles mainly related to cellular homeostasis and metabolism, which also have an effect on inflammatory response [16,17]. Recently, it has been shown that the overexpression of HNF1α increases the RelA/p65 mRNA level and decreases its degradation, leading to nuclear RelA/p65 accumulation and NF–κB activation in Huh7 cells [9]. Based on this and data indicating that (a) the promoter region of the gene encoding NF–κB (precisely, RelA/p65) possesses potential binding sites for HNF1α (AliBaba2.1) and (b) HNF1α is significantly elevated in the livers of CRF rats [18,19], we hypothesize that HNF1α could be involved in the upregulation of *RelA/p65* gene expression and, finally, in the activation of the NF–κB signaling pathway. In turn, the NF–κB signaling pathway could activate the transcription of various genes (i.e., NF–κB responsive genes), the products of which are tightly associated with inflammation and kidney function deterioration.

Peroxisome proliferator activated receptor α (PPAR α) is a ligand-activated transcriptional factor expressed in numerous organs including liver, heart, kidney and skeletal muscle [20]. It is activated by: (a) natural ligands such as fatty acids and synthetic compounds such as fibrates and (b) anti-lipidemic drugs [21]. Activated PPAR α regulates gene expression by heterodimerizing with retinoid X receptor (RXR) and binding to PPAR response elements [22]. In this way, fibrates (and other activators/agonists of PPAR α) stimulate catabolism of lipoproteins and expression of genes encoding enzymes involved in peroxisomal β-oxidation of fatty acids [23]. Numerous studies indicate that PPAR α also exerts direct anti-inflammatory effects [24,25]. It has been shown that fenofibrate inhibits myocardial inflammation (precisely, it decreases NF–κB activities and VCAM-1 and ICAM-1 expression in the heart) of angiotensin II infused rats [26]. Moreover, in an experimental model of myocardial infarction, PPAR α prevents overexpression of proinflammatory molecules including: IL-6, ICAM-1 and NF–κB [27]. The results from the above presented study indicate that in addition to their lipid-lowering properties, fibrates may also have beneficial effects in CKD patients by inhibiting inflammation. We have shown previously that clofibrate reduces HNF1α mRNA levels in the livers of CRF rats [18]. Therefore, it is likely that fibrates might decrease the biosynthesis of proinflammatory cytokines, chemokines and adhesion molecules via the inhibition of HNF1α.

Thus, we examined the relationship between RelA/p65 (the primary NF–κB member responsible for the transcriptional regulation of target genes) and *HNF1α* gene expression in the livers of CRF rats. To assess the direct impact of HNF1α on *NF–κB* gene expression, we examined the effect of small interfering RNA (siRNA) on the expression of the *HNF1α* gene and the *RelA/p65* gene in the HepG2 cell line. We also verified if an increase in *RelA/p65* gene expression was associated with changes in the NF–κB signaling pathway activity by measuring the expression of NF–κB responsive genes such as (a) interleukin-6 (*IL-6*), (b) vascular cell adhesion molecule 1 (*VCAM-1*), (c) intercellular cell adhesion molecule 1 (*ICAM-1*), and (d) monocyte chemoattractant protein 1 (*MCP-1*) in the livers of CRF rats. We also examined the effect of clofibrate, an anti-lipidemic drug that also lowers the serum concentration of IL-6, VCAM-1, ICAM-1, and MCP-1 [28] on HNF1α and RelA/p65 mRNA levels in the livers of CRF rats. 

The main purpose of the present study was to analyze the influence of HNF1α overexpression on *NF–κB* gene expression and the expression of genes responsive to the activation of the NF–κB signaling pathway, i.e., genes encoding IL-6, VCAM-1, ICAM-1, and MCP-1 in the livers of CRF rats—an experimental model of CKD in humans. Moreover we aimed to analyze the effect of inhibition of *HNF1α* gene expression by clofibrate on *NF–κB*, *IL-6*, *VCAM-1*, *ICAM-1*, and *MCP-1* gene expression.

## 2. Results

Mean serum creatinine concentrations, a marker of renal function found in CRF rats (2.7 mg/dL), were approximately four to five times higher than in sham-operated and pair-fed rats (0.57 and 0.60 mg/dL, respectively). Similarly, the mean serum blood urea nitrogen (BUN) concentration in CRF rats (267.6 mg/dL) was significantly higher than in control and pair-fed rats (51.9 and 50.6 mg/dL, respectively). These results validate our CRF rat experimental model and suggest that CRF induced by partial nephrectomy in rats corresponded, at a rough estimate, to the late stage of chronic kidney disease (CKD) in patients. 

To determine the role of HNF1α in the regulation of *RelA/p65* gene expression in the livers of CRF rats, we first examined the association between *RelA/p65* and *HNF1α* gene expression in CRF rats. Liver RelA/p65 mRNA levels were significantly higher in CRF than in the control or pair-fed rats (Figure 1A). The intergroup differences in liver RelA/p65 mRNA levels were reflected by the different serum NF–κB (RelA/p65) concentrations (Figure 1B). A strong positive correlation was found between serum creatinine concentration and liver RelA/p65 mRNA (Table 1), as well as between serum creatinine concentration and serum NF–κB concentrations (Table 1). Essentially, a similar relationship between serum BUN concentrations and liver RelA/p65 mRNA was observed (Table 1). This strong positive correlation between the liver RelA/p65 mRNA level and serum markers of renal function (Table 1) suggests that *RelA/p65* gene transcription is tightly related to the progression of kidney disease in an experimental model.

As expected, the upregulation of liver *RelA/p65* gene expression in CRF rats was tightly associated with an increase in liver HNF1α in both mRNA (Figure 2A) and protein levels (Figure 2B). Again, a strong positive correlation was found between serum creatinine (and BUN) concentrations and liver HNF1α mRNA (Table 1), as well as between serum creatinine (and BUN) concentration and liver HNF1α protein levels (Table 1). 

Together, the results presented in Figure 1 and Figure 2, as well as in Table 2, suggest that HNF1α overproduction, due to its gene overexpression, can promote the expression of the liver *RelA/p65* gene in CRF rats.

To verify whether HNF1α is involved in the upregulation of *RelA/p65* gene expression in the livers of CRF rats, we assessed the deregulation of *HNF1α* gene expression in HepG2 cells by silencing its expression using small interfering RNA (siRNA). The knockdown of the endogenous HNF1α expression measured as the HNF1α mRNA level (Figure 3A) by two siRNAs in HepG2 cells was accompanied by a decrease in the RelA/p65 mRNA level (Figure 3B). Both HNF1α and RelA/p65 mRNA inhibition reached approximately 60% at 48 h after cell transfection. These results indicate that *HNF1α* gene expression is involved in the regulation of liver *RelA/p65* gene expression. Thus, it is likely that the upregulation of *HNF1α* gene expression is directly related to the upregulation of *RelA/p65* gene expression in the livers of CRF rats. 

To obtain more information about the potential physiological significance of the increase in *RelA/p65* gene expression by HNF1α, we further examined the expression of certain genes responsive to the NF–κB signaling pathway in the livers of CRF rats. The results of our study indicate that liver IL-6 (Figure 4A), MCP-1 (Figure 5A), VCAM-1 (Figure 6A), and ICAM-1 (Figure 7A) mRNA levels were significantly higher in CRF rats than in control or pair-fed rats. Moreover, the intergroup differences in liver IL-6 (Figure 4A), MCP-1 (Figure 5A), VCAM-1 (Figure 6A), and ICAM-1 (Figure 7A) mRNA levels were reflected by different serum IL-6 (Figure 4B), MCP-1 (Figure 5B), VCAM-1 (Figure 6B), and ICAM-1 (Figure 7B) concentrations. As expected, the simultaneous upregulation of liver *HNF1α* and *RelA/p65* gene expression in CRF rats was tightly associated with an increase in liver IL-6, MCP-1, VCAM-1, and ICAM-1 mRNAs and serum concentrations (Table 2).

To confirm the hypothesis that HNF1α is involved in liver *IL-6*, *MCP-1*, *VCAM-1*, and *ICAM-1* gene upregulation in CRF rats through the increase in *RelA/p65* gene expression, we transfected HepG2 with siRNA targeting HNF1α. Silencing the endogenous HNF1α expression measured as the HNF1α mRNA level (Figure 8A) by two siRNAs in HepG2 cells was accompanied by a decrease in IL-6 (Figure 8B), MCP-1 (Figure 8C), VCAM-1 (Figure 8D), and ICAM-1 (Figure 8E) mRNA levels. Since RelA/p65 was downregulated in HNF1α-suppressed HepG2 (Figure 3), the changes in these cells presented in Figure 8 could also be partially caused by the inhibition of *RelA/p65* gene expression. These results suggest that *HNF1α* gene expression is actually involved in the regulation of liver IL-6, MCP-1, VCAM-1, and ICAM-1. Thus, it seems likely that the upregulation of *HNF1α* gene expression in CRF rats is directly related to upregulation of *RelA/p65* gene expression and, consequently, the activation of the NF–κB signaling pathway. In turn, this process leads to the overexpression of genes encoding IL-6, MCP-1, VCAM-1, and ICAM-1 in the livers of CRF rats (Figure 4, Figure 5, Figure 6 and Figure 7).

Fibrates reduced HNF1α mRNA levels in the livers of CRF rats [18] and downregulated the expression of proinflammatory molecules through the inhibition of the NF–κB signaling pathway in human endothelial cells and in an experimental model of myocardial infarction [24,29]. Correspondingly, we hypothesized that fibrates might decrease the biosynthesis of proinflammatory cytokines and chemokine and adhesion molecules via the inhibition of HNF1α, which decreases *RelA/p65* gene expression. To verify this hypothesis, we examined the effect of clofibrate on HNF1α, RelA/p65, Il-6, MCP-1, VCAM-1, and ICAM-1 mRNA in the livers of CRF rats treated with clofibrate. The results presented in Figure 9 indicate that clofibrate decreases HNF-1α, Il-6, MCP-1, VCAM-1, and ICAM-1 mRNA. In this experiment, we did not determine proteins levels encoded by corresponding genes. However, taking into account the results presented in Figure 1, Figure 2, Figure 4, Figure 5, Figure 6 and Figure 7, which indicate that intergroup differences in RelA/p65, HNF-1α, Il-6, MCP-1, VCAM-1, and ICAM-1 mRNA levels were reflected by different levels of RelA/p65, HNF-1α, Il-6, MCP-1, VCAM-1, and ICAM-1 protein level, it is likely that decreases in mRNA level caused by clofibrate are related to decreases in protein level. This further supports the view that HNF1α may play an important role in the regulation of RelA/p65 biosynthesis and IL-6, MCP-1, VCAM-1, and ICAM-1 in the experimental model of CKD in humans. 

## 3. Discussion

The aim of this study was to analyze the influence of HNF1α overexpression on *RelA/p65* (subunit of NF–κB) gene expression and the expression of certain genes responsive to the activation of the NF–κB signaling pathway in the livers of CRF rats—an experimental clinically relevant model of CKD. The most important finding presented in this paper is that the upregulation of *HNF1α* gene expression in the livers of CRF rats is tightly associated with RelA/p65 overexpression. This suggests that HNF1α, as a transcriptional factor, is involved in the upregulation of *RelA/p65* gene expression in the livers of CRF rats. This view was confirmed by: (a) a strong positive correlation between *RelA/p65* gene expression (measured as RelA/p65 mRNA level and serum NF–κB concentration) and *HNF1α* gene expression (determined as mRNA and liver protein levels) in CRF rats; (b) a coordinated reduction in the expression of *RelA/p65* and *HNF-1α* genes caused by clofibrate in CRF rats, and finally, (c) a significant decrease in RelA/p65 mRNA expression due to a *HNF1α* gene expression knockdown with siRNA in HepG2 cells. Moreover, gene encoding RelA/p65 possesses HNF1α potential binding sites in its promoter (Alibaba). Thus, the activation of *RelA/p65* gene transcription in the livers of CRF rats could result in HNF1α binding to the RelA/p65 promoter. Overall, our study demonstrates for the first time that the upregulation of *HNF1α* gene expression in CRF rats likely contributes to *RelA/p65* gene overexpression. Previously, it has been shown that the upregulation of *RelA/p65* gene expression is associated with the activation of the NF–κB signaling pathway in pancreatic cancer [8]. The results reported by Barma et al. [15] suggest that fatty acid-induced activation of the NF–κB signaling pathway is also associated with the overexpression of the *RelA/p65* gene in skeletal muscle cells, which in turn is linked to insulin resistance. Additionally, it has been reported that the overexpression of the *RelA/p65* gene in human colon cell lines [10] and in human breast cancer, M14 melanoma, and lung adenocarcinoma cell lines [7] is associated with the promotion of apoptosis. It is likely, therefore, that excess RelA/p65 pooling in the liver cells of CRF rats also leads to the NF–κB activation signaling pathway and consequently to the upregulation of responsive genes, including interleukin-6 (*IL-6*), vascular cell adhesion molecule 1 (*VCAM-1*), intercellular cell adhesion molecule 1 (*ICAM-1*), and monocyte chemoattractant protein 1 (*MCP-1*) in the livers of CRF rats. These results suggest that NF–κB could be a downstream component of the HNF1α-initiated signaling pathway in the livers of CRF rats. In this respect, our results resemble the data published by Lin et al. [9], who depicted that HNF1α overexpression in the human hepatoma cell line (Huh7 cells) leads to an increase in the RelA/p65 mRNA level, while the knockdown of *HNF1α* gene expression resulted in a significant decrease in RelA/p65 mRNA level. Moreover, they proved that the elevated level of RelA/p65 mRNA is associated with the nuclear accumulation of RelA/p65 and the activation of the NF–κB pathway [9]. It is likely, hence, that similar events occur in the livers of CRF rats. Thus, we postulate that the upregulation of HNF1α (Figure 2) leads to the overexpression of RelA/p65 (Figure 1) and the activation of the NF–κB signaling pathway. In turn, the activation of the NF–κB signaling pathway leads to greater expression of several responsive genes, such as *IL-6*, *VCAM-1*, *ICAM-1*, and *MCP-1* in the livers of CRF rats (Figure 4, Figure 5, Figure 6 and Figure 7). This is consistent with the results presented in Figure 8, which displays that endogenous *HNF1α* gene expression knockdown in HepG2 cells caused a significant decrease in all these molecules’ mRNA levels in the livers of CRF rats concomitantly with a decrease in RelA/p65 (Figure 3). This confirms that the production of proinflammatory cytokines, cell adhesion molecules, and chemokines involves, at least in part, the activation of the NF–κB signaling pathway through HNF1α. Thus, HNF1α seems to contribute to the inflammatory state by increasing in *RelA/p65* gene expression and consequently by activating the NF–κB pathway in CRF rats. It is tempting to speculate that the induction of *HNF1α* gene expression under conditions of persistent inflammatory stimuli or the dysfunctional regulatory mechanism that occurs in CKD may be a general mechanism leading to the activation of the NF–κB signaling pathway and to the transcriptional induction of several genes involved in the progression of CKD and some related diseases. Nevertheless, further research is needed to investigate: (a) the molecular mechanism involved in the upregulation of *RelA/p65* gene expression by HNF-1α and (b) the role of HNF1α in the upregulation of NF–κB in patients with CKD.

In contrast to our data and Lin et al. [9], Qian et al. [30] observed that HNF1α repression in primary hepatocytes isolated from the livers of rats treated with dimethylnitrosamine leads to the activation of NF–κB. Moreover, Bao et al. [31] reported that the NF–κB signaling pathway downregulates HNF1α via an inhibition of miR-194 in hepatocellular carcinoma. These findings suggest that HNF1α and NF–κB under some conditions (for instance, the treatment of cells with dimethylnitrosamine or other pathological conditions) could play an opposite role in the inflammatory response. However, this contrasts with the commonly accepted view that both HNF1α [16] and NF–κB [6] serve as inflammatory mediators. Dimethylnitrosamine, a hepatic fibrosis-inducing agent [32], exerts a deep impact on cell function and finally induces hepatocarcinogenesis [33]. Moreover, in patients with hepatocellular carcinoma, lower *HNF4 α* gene expression was observed [34]. Therefore, one can assume that metabolic dysregulation by dimethylnitrosamine and cancer development could be one possible explanation for this discrepancy between our data, the data reported by Lin et al. [9], and results reported by Qian et al. [30] or by Bao et al. [31].

Peroxisome proliferator activated receptor α (PPARα) is a ligand-activated transcription factor that regulates gene expression by forming a heterodimer with retinoid X receptor (RXR) and binds to the PPAR response element present in several genes [22]. PPARα is highly expressed in the liver [20], is activated by natural (for instance, fatty acids and its metabolites) and synthetic (such as fibrates) ligands [21], and exerts anti-inflammatory activity [25]. Previously, it has been illustrated that fibrates (PPARα agonist) might inhibit inflammatory responses in cells by suppressing RelA/p65, a subunit of the NF–κB signaling pathway [35]. The results presented in this paper indicate that clofibrate can interfere negatively with HNF1α, RelA/p65, VCAM-1, ICAM-1, MCP-1, and IL-6 mRNA levels in the livers of CRF rats (Figure 9). Thus, our results agree with previous reports depicting that the activation of PPARα by clofibrate results in the inhibition of the NF–κB signaling pathway that regulates the expression of genes encoding ICAM-1 and VCAM [36,37,38]. Given that HNF1α and RelA/p65 are coordinately downregulated by clofibrate in the livers of CRF rats (Figure 9), it is likely that in our experimental model of CKD, clofibrate inhibits *HNF1α* gene expression, leading to the downregulation of genes encoding RelA/p65 and the inhibition of the NF–κB signaling pathway, which finally downregulates the expression of the studied adhesion molecules, chemokines, and cytokines (Figure 9). Overall, it seems that clofibrate may lead (via the activation of PPARα) to the downregulation of *HNF1α* gene expression, which modulates adhesion molecules, chemokines, and cytokines via the inhibition of transcription factors RelA/p65. Whether clofibrate directly inhibits the transcription of the *HNF1α* gene by the activation of PPARα remains to be verified. Notably, the transcriptional suppression of HNF-4 in the livers of rats treated with bezafibrate has also been reported by Hertz et al. [39]. Therefore, our results suggest that the *HNF1α* (such as *HNF4*) gene can be downregulated by fibrates, which leads to the suppression of target genes such as the *p65/RelA* subunit of the NF–κB signaling pathway. However, we cannot exclude that another mechanism (besides the inhibition of *HNF1 α* gene expression) could be involved in the regulation of *RelA/p65* gene expression by clofibrate.

This is the novel proposal that the anti-inflammatory effect of clofibrate is due, at least in part, to the inhibition of *HNF1α* gene expression. Therefore, it appears plausible that pharmacological approaches to block HNF1α upregulation may be beneficial to decrease the level of NF–κB, which may prevent an inflammatory state. 

The results presented in this paper suggest that the overexpression of HNF1α in the livers of CRF rats resulted in the increase in *RelA/p65* gene expression. This is associated with the increased expression of genes encoding IL-6, VCAM-1, ICAM-1, and MCP-1. Accordingly, it is likely that the overexpression of HNF1α leads to an increase in *RelA/p65* gene expression and the NF–κB activation pathway, which finally upregulates the expression of adhesion molecules, chemokines, and cytokines in CRF rats. Moreover, we showed that clofibrate decreases the gene expression of *IL-6*, *VCAM-1*, *ICAM-1*, and *MCP-1* in the liver, probably through the inhibition of *HNF-1α* and *RelA/p65* genes’ expression in the livers of CRF rats. 

## 4. Materials and Methods

### 4.1. Experimental Model of CKD

The studied model of advanced CKD was induced by subtotal nephrectomy as described previously [40]. Sham-operated animals served as the control. All animals (10 rats in each group, i.e., CRF; sham-operated and pair-fed rats) were kept in individual wire-mesh cages and allowed free access to tap water. CRF and sham-operated rats were allowed free access to a commercial diet that has been previously described [41]. Pair-fed rats received the same amount of food as consumed by CRF ones. Air temperature in the animal room was set at 22 °C, and the lighting schedule was controlled (12 h light/dark cycles). Six weeks after induction of CRF (between 8.00 and 10.00 am): (a) blood samples from abdominal aorta, and (b) pieces (approximately 0.5 g) of liver were collected under thiopental anesthesia. Then, rats were euthanized. Pieces of liver were rapidly frozen in liquid nitrogen and then stored at −80 °C until the expression of the studied genes was determined. Serum was obtained after blood centrifugation at 1500× *g* for 10 min.

### 4.2. Clofibrate Treatment of Rats with CRF

Five weeks after the induction of CRF, the rats were given clofibrate (250 mg/kg of body weight for seven successive days) as described previously [42]. 

### 4.3. HepG2 Cell Culture

HepG2 cells (a human hepatocellular carcinoma cell line) were obtained from ATCC (Manassas, VA, USA). Cells were maintained in standard Minimum Essential Eagle’s Medium (M5650; Sigma, 3050 Spruce Street, St. Luis, MO, USA) supplemented with: 2 mM glutamine, 10% fetal bovine serum, penicillin (100 IU per mL), and streptomycin (100 µg per mL) at 37 °C under a humidified 95%/5% (vol/vol) mixture of air and CO_2_. Two days before the main experiments, HepG2 cells were passaged in 6-well plates at 10 × 10^−4^ cells per well. Then, cells were cultured and grown to approximately 70% confluence.

### 4.4. Small Interfering RNA (siRNA) Transfection

Two different sequences of siRNA targeting HNF-1α were used: (a) Hs-TCF1-2, no SI00011620, and (b) Hs-TCF1-5, no SI03095015. All Stars Negative Control, no. 1027280 was used as negative control (siRNA NC). siRNAs were obtained from Qiagen (Crawley, UK). HepG2 cells treated by lipofectamine were used as a control. HepG2 cells were transfected with siRNA at concentrations of 10 nM using 0.1% (*v/v*) Lipofectamine RNA iMAX (Invitrogen, Paisley, UK), as described in the manufacturer’s protocol. Transfection was performed in serum-free OptiMEM (Invitrogen, Paisley, UK). After 48 h, cells were harvested and used for total RNA or protein extraction (see below).

### 4.5. Liver and HepG2 Cells RNA Isolation

Total RNA was extracted from the frozen liver using the guanidinium isothiocyanate–phenol/chloroform method [43]. GenElute™ Mammalian Total RNA Miniprep Kit (Sigma, 3050 Spruce Street, St. Luis, MO, USA) was used for isolation of total RNA from HepG2 cells. The obtained RNA concentration was determined from the absorbance at 260 nm. All obtained samples had a 260/280 nm absorbance ratio of about 2.0.

### 4.6. cDNA Synthesis

First-strand cDNA was synthesized from 1 µg of total RNA (RevertAid First Strand cDNA Synthesis Kit, Thermo Fisher Scientific, V. A. Graiciuno 8, Vilnius, Lithuania). Prior to amplification of cDNA, each RNA sample was treated with RNase-free DNase I (Thermo Fisher Scientific, V. A. Graiciuno 8, Vilnius, LT-02241 Lithuania) at 37 °C for 30 min.

### 4.7. Determination of HNF1, p65/RelA, MPC-1, VCAM-1, ICAM-1 mRNA Level by RT-PCR

Rat HNF1α, p65/RelA, MPC-1, VCAM-1, ICAM-1, β-actin, and TBP (TATA-box binding protein) mRNA levels were quantified by RT-PCR using a Chromo4 Real Time Detection System (Bio-Rad Laboratories, 1000 Alfred Nobel Drive Hercules, CA, USA). Primers were designed with Sequence Analysis software package (Informagen, Newington, 375 Little Bay Rd, Newington, NH, USA) from gene sequences obtained from the Ensembl Genome Browser (www.ensembl.org, accessed on 16 January 2018). The rat sequences of primer pairs (sense and antisense) used in this study are: (a) 5′-AAGATGACACGGATGACGATGG-3′ (sense) and 5′-GGTTGAGACCCGTAGTGTCC-3′ (antisense) for the HNF1α; (b) 5′-ACACATAGCGGCTGGAAGA-3′ (sense) and 5′-TTCTCCACCAGGGGGTCT-3′ (antisense) for the p65/RelA; (c) 5′-TCCTACCCCAACTTCCAATGCTC -3′ (sense) and 5′-TTGGATGGTCTTGGTCCTTAGCC-3′ (antisense) for IL-6; (e) 5′-GCTGCTACTCATTCACTGGC-3′ (sense) and 5′-GCTGCTACTCATTCACTGGC-3′ (antisense) for the MCP-1; (e) 5′-CTTCGCTGACAGGTCACAGT-3′ (sense) and 5′-GTTGGCTGTGACTCTCCCTC-3′ (antisense) for the VCAM-1; (f) 5′-CGCCAGAGGAAGATCAGGAT-3′ (sense) and 5′-AGGTGGGTGAGGGGTAAATG-3′ (antisense) for the ICAM-1; (g} 5′-TGTCACCAACTGGGACGATA-3′ (sense) and 5′ GGGGTGTTGAAGGTCTCAAA-3′ (antisense) for β-actin; (h) 5′-CACCGTGAATCTTGGCTGTAAAC-3′ (sense) and 5′-ATGATGACTGCAGCAAACCG-3′ (antisense) for the *Tbp.*


Primers for human: (a) HNF1α (qHsaCED0001918), (b) NF–κB (qHsaCED0002379), (c) IL-6 (qHsaCID0020314), (d) MCP-1(qHsaCID0011608), (e) VCAM-1 (qHsaCID0016779), (f) ICAM-1 (qHsaCED0004281), (g) β-actin (qHsaCED0036269), and (h) TBP (qHsaCID0007122) assayed in HepG2 cells were obtained from Bio-Rad Laboratories, Inc., Hercules, CA, USA. Real-time PCR amplification was performed in 20 μL volumes using iQ SYBR Green Supermix (Bio-Rad Laboratories, Hercules, CA, USA). Each reaction contained cDNA and 0.3 μM of each primer. Control reactions, with omission of the RT step or with no template cDNA added, were performed with each assay. All samples were run in triplicate. To compensate for variations in the amount of added RNA and in the efficiency of the reverse transcription, β-actin or TBP mRNA was quantified in the corresponding samples, and the results were normalized to these values. Results obtained with β-actin and TBP (as internal standards) were similar. The relative quantities of transcripts were calculated using the 2^−ΔΔCT^ formula [44]. The results are expressed in arbitrary units, with one unit representing the mean mRNA level determined in a control group. The amplification of specific transcripts was confirmed by obtaining the melting curve profiles and subjecting the amplification products to agarose gel electrophoresis.

### 4.8. Western Blot Analysis of HNF1α and β-Actin in Rat Liver

Frozen liver samples were homogenized in a buffer containing 10 mM Tris–HCl (pH 7.8), 2% SDS, 10 mM DTT, and proteinase inhibitors (Sigma) and centrifuged at 15,000× *g* for 20 min at 20 °C. Supernatants were collected, and the protein concentration was determined by Bradford assay. Tissue lysates containing 20 μg (liver) of total protein were separated by 10% SDS-PAGE and electroblotted onto Immobilon Transfer Membrane (Millipore Corporation, Billerica, MA, USA). The following antibodies were used: monoclonal antibody against HNF-1 (sc-393925, Santa Cruz Biotechnology, Inc. 10410 Finnell Street Dallas, TX, USA) and polyclonal antibody against Actin (sc-7210, Santa Cruz Biotechnology). HRP-conjugated secondary antibodies (sc-2030 and sc-2004) were obtained from Santa Cruz Biotechnology and the HAF019 from R&D Systems, Minneapolis, MN, USA) Immunodetection was accomplished with enhanced chemiluminescence using Western blotting Luminol Reagent (sc-2048, Santa Cruz Biotechnology add the location of company).

### 4.9. Determination of HNF1, p65/RelA, MPC-1, VCAM-1, ICAM-1 Concentration in Serum of CRF Rats

Commercially available ELISA kits were used to estimate proteins in serum concentration: (a) NF–κB (RelA/p65) (antibodies-on line GmbH, Aachen, Germany); (b) interleukin-6 (IL-6) (R&D Systems, Minneapolis, MN, USA); (c) monocyte chemoattractant protein-1 (MCP-1/CCL2) (R&D Systems, Minneapolis, MN, USA); (d) vascular cell adhesion molecule-1 (VCAM-1/CD106) (elabscience, Houston, TX, USA); (e) intercellular cell adhesion molecule-1 (ICAM-1/CD54) (R&D Systems, Minneapolis, MN, USA).

### 4.10. Serum Creatinine and Blood Urea Nitrogen (BUN) Concentration

Serum creatinine and BUN concentrations were determined using a Hitachi 704 auto analyzer.

### 4.11. Statistics

The statistical significance of differences between groups was assessed by one-way analysis of variance (ANOVA) followed by Student’s *t* test and one-way analysis of variance (ANOVA), followed by Tukey’s post hoc test. The Sigma Stat software was used. The results are presented as mean ± SD. Differences between groups were considered significant when *p* < 0.05. The relations between two variables were calculated using Pearson’s correlation.

## 5. Conclusions

The results presented herein for the first time provide new information about (a) the role of HNF1α in the upregulation of genes encoding RelA/p65 and (b) the activation of NF–κB, which was reflected by the overexpression of genes responsive to NF–κB in CRF rats. Moreover, we found that the inhibition of *HNF1α* gene expression by clofibrate led to a significant decrease in RelA/p65 and its target gene expression. 

Our findings suggest that NF–κB could be a downstream component (element) of the HNF1α-initiated signaling pathway in liver of chronic renal failure rats. This helps to understand better pathology of chronic kidney disease in humans and identifies potential new therapeutic targets. 

## Figures and Tables

**Figure 1 ijms-23-08883-f001:**
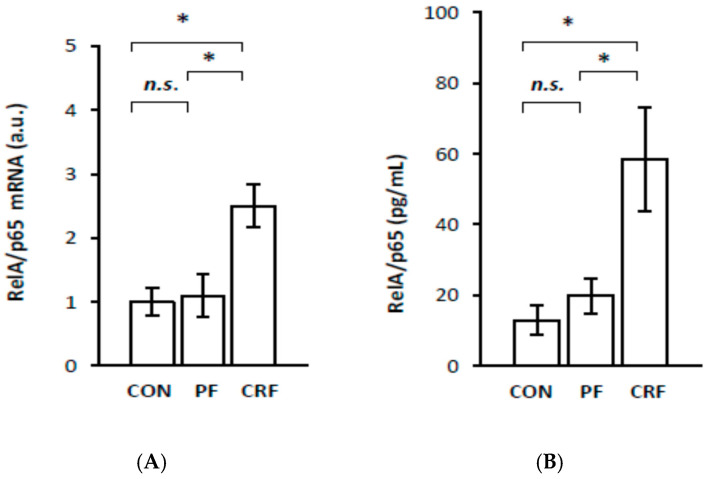
RelA/p65 mRNA level in liver (**A**) and serum concentration of RelA/p65 (**B**) in control (CON), pair-fed (PF), and chronic renal failure (CRF) rats. Graphs represent the mean ± SD from 10 controls, 10 pair-fed and 10 chronic renal failure rats. Statistics: * *p* < 0.05, *n.s.* (not significant}.

**Figure 2 ijms-23-08883-f002:**
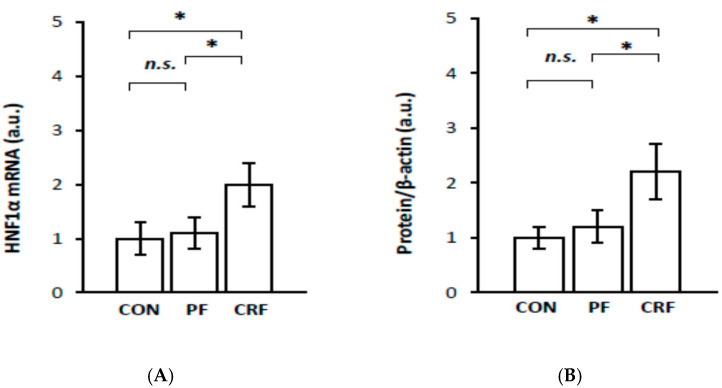
Hnf1α mRNA level in liver (**A**) and densitometric analysis of Western blot bands of liver HNF1α protein level (**B**) in control (CON), pair-fed (PF), and chronic renal failure (CRF) rats. Graphs represent the mean ± SD from 10 controls, 10 pair-fed and 10 chronic renal failure rats. Statistics: * *p* < 0.05, *n.s.* (not significant).

**Figure 3 ijms-23-08883-f003:**
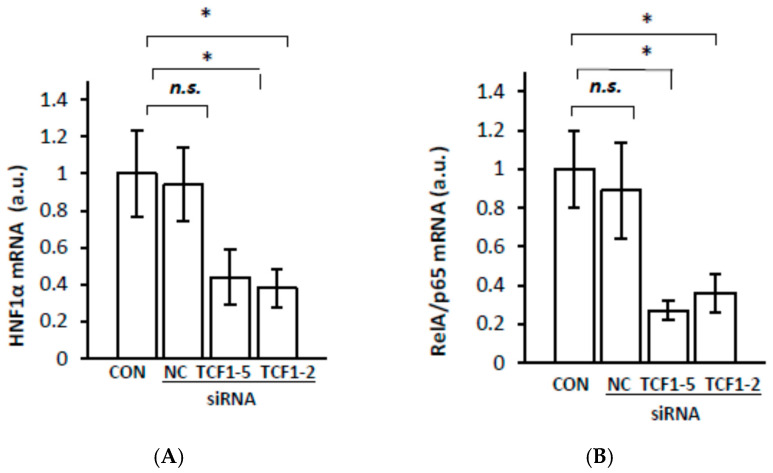
Coordinated inhibition of *HNF1α* (**A**), *RelA/p65* (**B**) expression in HepG2 cells by two different sequences of siRNA targeting *HNF-1α*: lipofectamine treated HepG2 cells (CON), cells transfected with siRNA targeting *HNF-1α* (TCF1-2 or TCF1-5) or negative control (NC). Graphs represent the mean (±SD) from 6 plates performed in three different experiments. Statistics: * *p* < 0.05, *n.s.* (not significant).

**Figure 4 ijms-23-08883-f004:**
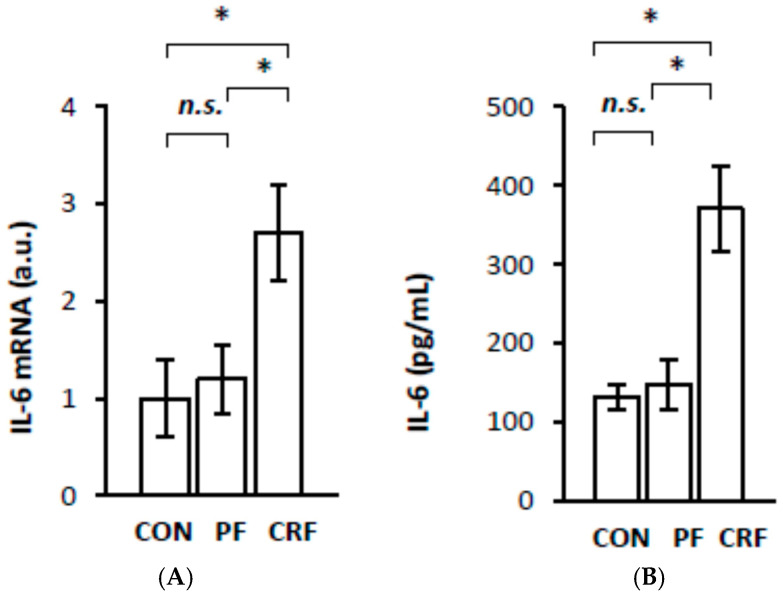
IL-6 mRNA level in liver (**A**) and serum IL-6 concentration (**B**) in control (CON), pair-fed (PF), and chronic renal failure (CRF) rats. Graphs represent the mean ± SD from 10 controls, 10 pair-fed and 10 chronic renal failure rats. Statistics: * *p* < 0.05, *n.s.* (not significant).

**Figure 5 ijms-23-08883-f005:**
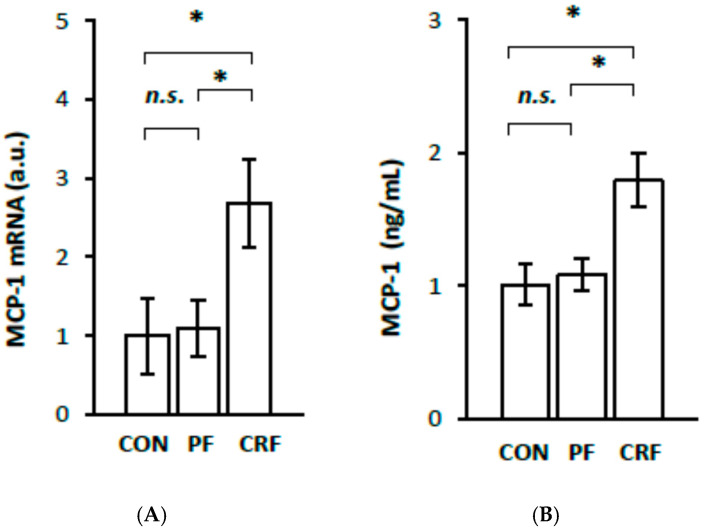
MCP-1 mRNA level in liver (**A**) and serum MCP-1 concentration (**B**) in control (CON), pair-fed (PF), and chronic renal failure (CRF) rats. Graphs represent the mean ± SD from 10 controls, 10 pair-fed and 10 chronic renal failure rats. Statistics: * *p* < 0.05, *n.s.* (not significant).

**Figure 6 ijms-23-08883-f006:**
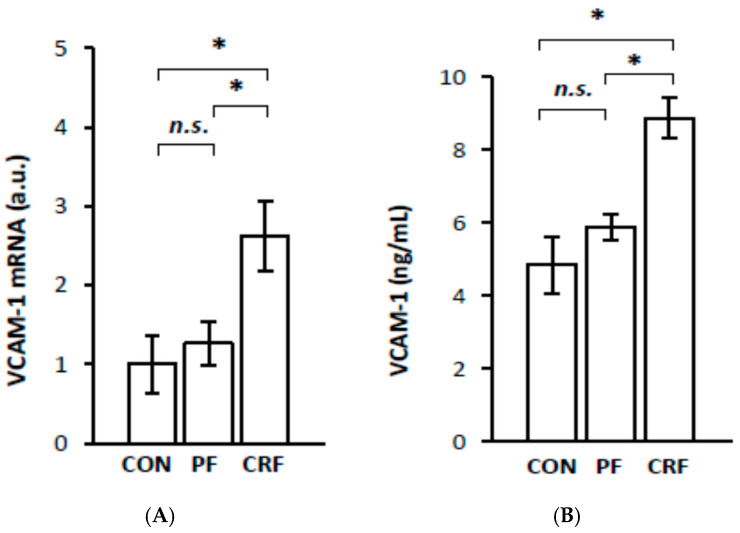
VCAM-1 mRNA level in liver (**A**) and serum VCAM-1 concentration (**B**) in control (CON), pair-fed (PF), and chronic renal failure (CRF) rats. Graphs represent the mean ± SD from 10 controls, 10 pair-fed and 10 chronic renal failure rats. Statistics: * *p* < 0.05, *n.s.* (not significant).

**Figure 7 ijms-23-08883-f007:**
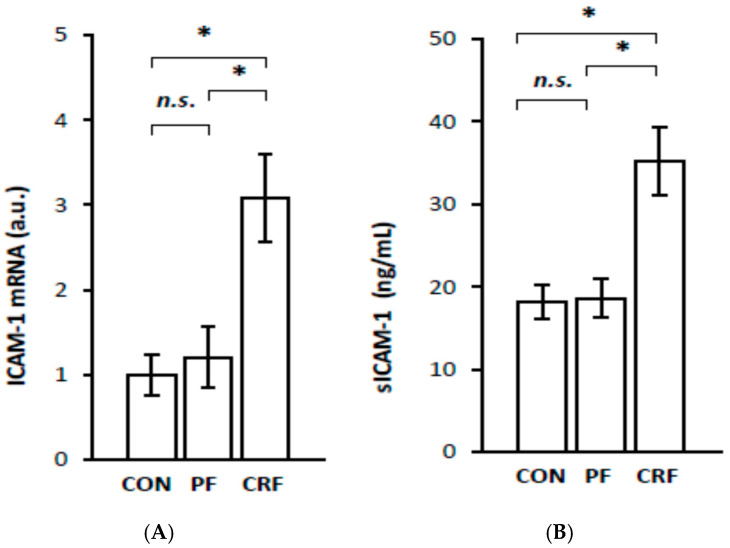
ICAM-1 mRNA level in liver (**A**) and serum sICAM-1 concentration (**B**) in control (CON), pair-fed (PF), and chronic renal failure (CRF) rats. Graphs represent the mean ± SD from 10 controls, 10 pair-fed and 10 chronic renal failure rats. Statistics: * *p* < 0.05, *n.s.* (not significant).

**Figure 8 ijms-23-08883-f008:**
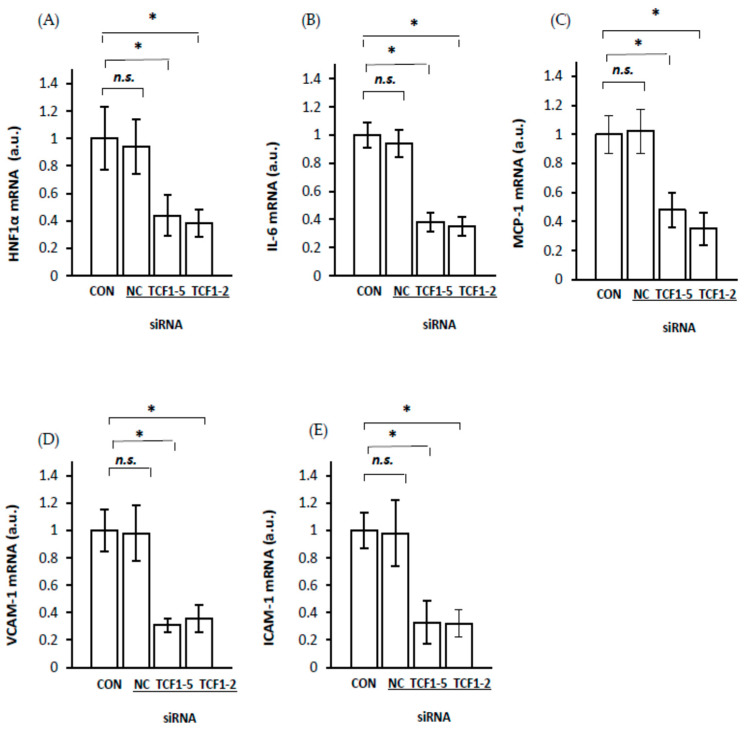
Coordinated inhibition of *HNF1α* (**A**), *IL-6* (**B**), *MCP-1* (**C**), *VCAM-1* (**D**), *ICAM* (**E**) expression in HepG2 cells by two different sequences of siRNA targeting *HNF-1α*: lipofectamine-treated HepG2 cells (CON), cells transfected with siRNA targeting *HNF-1α* (TCF1–2 or TCF1–5) or negative control (NC). Graphs represent the mean ± SD of results from 6 plates performed in three different experiments. Statistics: * *p* < 0.05, *n.s.* (not significant).

**Figure 9 ijms-23-08883-f009:**
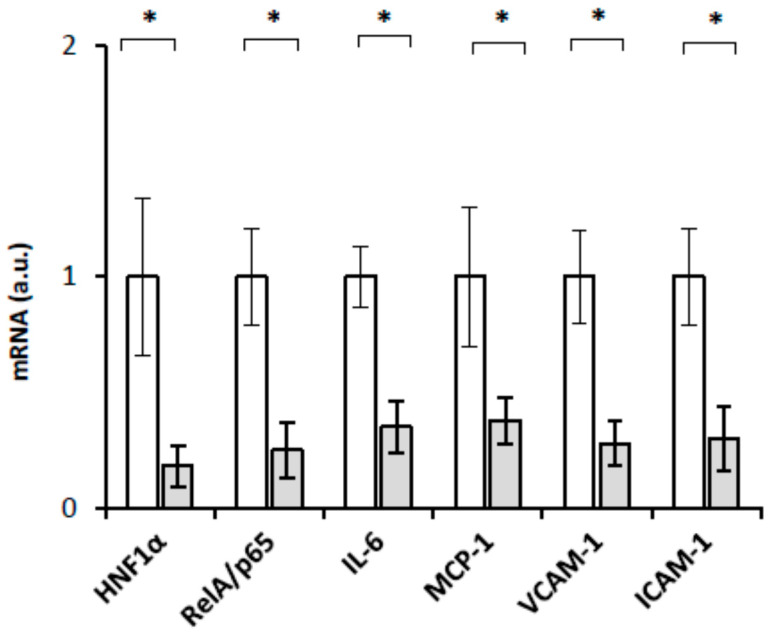
Relative HNF1α, NF–κB, IL-6, MCP-1, VCAM-1, ICAM mRNA levels in liver of untreated (□) and clofibrate-treated (
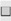
) rats with chronic renal failure. Graphs represent the mean ± SD of results from 10 rats. β-actin and TBP mRNA was quantified in the corresponding samples, and the results regarding mRNA levels were normalized to these values (a.u., arbitrary units); * *p* < 0.05.

**Table 1 ijms-23-08883-t001:** Correlation coefficients of serum creatinine concentrations or BUN with NF1α, NF–κB, IL-6, MCP-1, VCAM-1 or ICAM-1 expression or protein concentration.

Studied Parameters		Creatinine	BUN
NF1α	mRNA	0.80 **	0.73 **
	protein	0.84 **	0.82 **
NF–κB	mRNA	0.78 **	0.84 **
	protein	0.83 **	0.89 **
IL-6	mRNA	0.72 **	0.76 **
	protein	0.77 **	0.76 **
MCP-1	mRNA	0.97 **	0.88 **
	protein	0.74**	0.80 **
VCAM-1	mRNA	0.92 **	0.89 **
	protein	0.69 *	0.77 **
ICAM-1	mRNA	0.97 **	0.93 **
	protein	0.87 **	0.82 **

Statistics: * *p*< 0.05; ** *p*< 0.001.

**Table 2 ijms-23-08883-t002:** Coefficient of correlation between HNF1α mRNA or protein and studied parameters.

Studied Parameters		HNF1α mRNA	HNF1α Protein
NF–κB	mRNA	0.72 **	0.88 **
	protein	0.63 *	0.72 **
IL-6	mRNA	0.77 **	0.74 **
	protein	0.67 **	0.83 **
MCP-1	mRNA	0.92 **	0.74 **
	protein	0.82 **	0.65 *
VCAM-1	mRNA	0.81 **	0.71 **
	protein	0.73 *	0.83 **
ICAM-1	mRNA	0.87 **	0.74 **
	protein	0.93 **	0.76 **

Statistics: * *p*< 0.05; ** *p* < 0.001.

## Data Availability

Not applicable.

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
