# Peer review of "Hepatocyte Nuclear Factor 1α Proinflammatory Effect Linked to the Overexpression of Liver Nuclear Factor–κB in Experimental Model of Chronic Kidney Disease"

_ijms, 2022, doi:10.3390/ijms23168883_

Round 1
Reviewer 1 Report
The manuscript entitled: "HNF1α proinflammatory effect linked to the overexpression ..." is basically a very good work that requires visual rather than substantive correction.
However, this has to be done before publishing.
Below are my comments.
1. Please edit the title, rather than unnecessary names, and abbreviations, it looks strange.
2. In keywords, instead of abbreviations, I would make more general statements.
3. I would greatly expand the introduction, necessarily. There is really a lot to write about with such topics. It is enough to enter the keywords proposed by the authors into the browser to have a lot to choose from.
4. In the last paragraph of the introduction, please also specify the purpose of the research more clearly. And the last paragraph is more about conclusions, so it either needs to be edited or moved to conclusions.
5. Fig 1 - To be corrected, no error bars are visible. They are probably the same color as the data. Such a presentation is unacceptable.
6. Same in Fig. 2. Besides, the mark (n.s.) is blurred.
7. Figs 3 and 4 and 5 and 6 also to be improved - same as Figs 1 and 2.
8. Fig 8 also to be improved. In panel A you can see a black line in the sense of a triangle - what is it?
9. Fig. 9 - I think you know what I mean.
10. The authors wrote in the conclusions section. Please complete this. Move the last paragraph from the Introduction to the conclusions. Necessarily.
11. Literature items can be gently supplemented.
Overall a very nice job. It is a pity that the form of the presentation is a bit underdeveloped. But these are minor mistakes of course. And they do not affect the merits. I mark Major Revision but only by the presentation. Because the content of the manuscript is at a high level. I greet the authors from Poland.
After the corrections, I would like to see the work again.
Author Response
Author’s responses to Reviewer 1 comments
We are greatly grateful to you for taking time to read this manuscript and for pointing out all constructive suggestions. According to these suggestions, we made the following changes in the manuscript.
1. Please edit the title, rather than unnecessary names, and abbreviations, it looks strange.
Response: The title has been edited
2. In keywords, instead of abbreviations, I would make more general statements.
Response: According to your suggestion keywords has been changed
3. I would greatly expand the introduction, necessarily. There is really a lot to write about with such topics. It is enough to enter the keywords proposed by the authors into the browser to have a lot to choose from.
Response: Introduction has been expanded
4. In the last paragraph of the introduction, please also specify the purpose of the research more clearly. And the last paragraph is more about conclusions, so it either needs to be edited or moved to conclusions.
Response: In the last paragraph of the introduction, the purpose of the research has been specified. Moreover, the last paragraph from the former version has been moved to conclusion of the present version of the manuscript.
5. Fig 1 - To be corrected, no error bars are visible. They are probably the same color as the data. Such a presentation is unacceptable.
Response: Figure 1 has been corrected.
6. Same in Fig. 2. Besides, the mark (n.s.) is blurred.
Response: Figure 2 has been corrected.
7. Figs 3 and 4 and 5 and 6 also to be improved - same as Figs 1 and 2.
Response: Figures 3, 4, 5 and 6 has been corrected.
8. Fig 8 also to be improved. In panel A you can see a black line in the sense of a triangle - what is it?
Response: Figure 8 has been improved.
9. Fig. 9 - I think you know what I mean.
Response: Figure 9 has been improved.
10. The authors wrote in the conclusions section. Please complete this. Move the last paragraph from the Introduction to the conclusions. Necessarily.
Response: This has been done
11. Literature items can be gently supplemented.
Response: Literature (20-27) was supplemented/
Finally allow us to express our gratitude to you for your valuable comments and suggestions on this manuscript.
Reviewer 2 Report
The authors described the discovery of coordinated overexpression of genes encoding RelA/p65 and HNF1α in the liver of CRF mice.
The experiment presented by the authors is the result of checking the gene level, so verification at the protein level is necessary.
The authors use liver tissue from rats that induced renal failure to identify the protein expression that they consider most important at the protein level and add to the results.
Author Response
Author’s responses to Reviewer 2 comments
We are greatly grateful to you for taking time to read this manuscript and for pointing out all constructive suggestions. According to these suggestions, we made the following changes in the manuscript.
The authors described the discovery of coordinated overexpression of genes encoding RelA/p65 and HNF1α in the liver of CRF mice.
The experiment presented by the authors is the result of checking the gene level, so verification at the protein level is necessary.
Response: In the most experiments presented in this paper we measured and presented (see Figures 1, 2, 4, 5, 6, 7) both mRNA and proteins levels encoded by studied genes. Moreover, we observed strong positive correlation between mRNA levels and protein levels encoded by studied genes.
The authors use liver tissue from rats that induced renal failure to identify the protein expression that they consider most important at the protein level and add to the results.
Response: Given that strong positive correlation between mRNA levels and protein levels encoded by studied genes (see above), we believe that significant decrease in mRNA levels caused by clofibrate (Figure 9) are related to decrease in protein levels. This was mentioned in the text (page 11; lines 367-372) as follow: (…) In this experiment we did not determine proteins levels encoded by corresponding genes. However taking into account the results presented on Figures 1, 2, 4, 5, 6, 7, which indicate that intergroup differences in RelA/p65, HNF-1α, Il-6 , MCP-1, VCAM-1, and ICAM-1 mRNA levels were reflected by different levels of RelA/p65, HNF-1α, Il-6 , MCP-1, VCAM-1, and ICAM-1 protein level, it is very likely that decrease in mRNA level cause by clofibrate is tightly related to decrease in protein level (…).
Round 2
Reviewer 1 Report
The authors addressed all of the previous queries, therefore the manuscript may be accepted in the present form.
Reviewer 2 Report
The authors' revised manuscripts are sufficient to enhance the understanding of the research topic.